# COVID-19 CG enables SARS-CoV-2 mutation and lineage tracking by locations and dates of interest

**Albert Tian Chen[1], Kevin Altschuler[2], Shing Hei Zhan[3,4†], Yujia Alina Chan[1†], Benjamin E Deverman[1†]***

[1]Stanley Center for Psychiatric Research, Broad Institute of MIT and Harvard, Cambridge, United States; [2]Independent web designer, Cambridge, United States; [3]Department of Zoology & Biodiversity Research Centre, the University of British Columbia, Vancouver, Canada; [4]Fusion Genomics Corporation, Burnaby, Canada

**Abstract** COVID-19 CG (covidcg.org) is an open resource for tracking SARS-CoV-2 single-nucleotide variations (SNVs), lineages, and clades using the virus genomes on the GISAID database while filtering by location, date, gene, and mutation of interest. COVID-19 CG provides significant time, labor, and cost-saving utility to projects on SARS-CoV-2 transmission, evolution, diagnostics, therapeutics, vaccines, and intervention tracking. Here, we describe case studies in which users can interrogate (1) SNVs in the SARS-CoV-2 spike receptor binding domain (RBD) across different geographical regions to inform the design and testing of therapeutics, (2) SNVs that may impact the sensitivity of commonly used diagnostic primers, and (3) the emergence of a dominant lineage harboring an S477N RBD mutation in Australia in 2020. To accelerate COVID-19 efforts, COVID-19 CG will be upgraded with new features for users to rapidly pinpoint mutations as the virus evolves throughout the pandemic and in response to therapeutic and public health interventions.

**\*For correspondence:**
bdeverma@broadinstitute.org

†These authors contributed equally to this work

## Introduction

Since the beginning of the pandemic, SARS-CoV-2 genomic data has been accumulating at an unprecedented rate (400,000+ virus genomes as of February 2020 on the GISAID database) (*Elbe and Buckland-Merrett, 2017*; *Shu and McCauley, 2017*). Numerous countries have mobilized to sequence thousands of SARS-CoV-2 genomes upon the occurrence of local outbreaks, collectively and consistently contributing more than 20,000 genomes per month (*Figure 1—figure supplement 1A,B*). It is important to note that, despite the slow accumulation of potentially functional (nonsynonymous) mutations, there has been a steady increase in the number of variants with more than six nonsynonymous mutations compared to the WIV04 reference, an early genome of SARS-CoV-2 that was sampled in Wuhan in December 2019 (*Figure 1—figure supplement 1C*). To evaluate the outcomes of anti-COVID-19 measures and detect keystone events of virus evolution, it is important to track changes in SARS-CoV-2 mutation and population dynamics in a location and date-specific manner. Indeed, several countries and the National Institutes of Health (NIH) have recognized how critical it is to collect SARS-CoV-2 genomic data to support contact tracing efforts and to inform public health decisions – these are paramount to the re-opening of countries and inter-regional travel (*Collins, 2020*; *Gudbjartsson et al., 2020*; *Oude Munnink et al., 2020*; *Virological, 2020*; *Rockett et al., 2020*). Yet, the quantity and complexity of SARS-CoV-2 genomic data (and metadata) make it challenging and costly for the majority of scientists to stay abreast of SARS-CoV-2 mutations in a way that is meaningful to their specific research goals. Currently, each group or organization has to independently expend labor, computing costs, and, most importantly, time to curate and analyze

**eLife digest** The discovery of faster spreading variants of the virus that causes coronavirus disease 2019 (COVID-19) has raised alarm. These new variants are the result of changes (called mutations) in the virus' genetic code. Random mutations can occur each time a virus multiplies. Although most mutations do not introduce any meaningful changes, some can alter the characteristics of the virus, for instance, helping the virus to spread more easily, reinfecting people who have had COVID-19 before, or reducing the sensitivity to treatments or vaccines.

Scientists need to know about mutations in the virus that make treatments or vaccines less effective as soon as possible, so they can adjust their pandemic response. As a result, tracking these genetic changes is essential. But individual scientists or public health agencies may not have the staff, time or computer resources to extract usable information from the growing amount of genetic data available.

A free online tool created by Chen et al. may help scientists and public health officials to track changes to the virus more easily. The COVID-19 CoV Genetics tool (COVID-19 CG) can quickly provide information on which virus mutations are present in an area during a specific period. It does this by processing data on mutations found in viral genetic material collected worldwide from hundreds of thousands of people with COVID-19, which are hosted in an existing online database. The COVID-19 CG tool presents customizable, interactive visualizations of the data.

Thousands of scientists, public health agencies, and COVID-19 vaccine and treatment developers in over 100 countries are already using the COVID-19 CG tool to find the most common mutations in their area and use it for research. They can use this information to develop more effective vaccines or treatments. Chen et al. plan to update and improve the tool as more information becomes available to help advance global efforts to end the COVID-19 pandemic.

the genomic data from GISAID before they can generate specific hypotheses about SARS-CoV-2 lineages and mutations in their population(s) of interest.

## Results

To address this challenge, we built COVID-19 CoV Genetics (COVID-19 CG, covidcg.org), a performant, interactive, and fully scalable web application that tracks SARS-CoV-2 single-nucleotide variants (SNVs), lineages, and clades without sub-sampling. COVID-19 CG is a free, open access interface that allows users to adapt analyses according to their dates and locations of interest (*Figure 1A,B*; data processing workflow in *Figure 1—figure supplement 2*). Users can also select and compare trends in SARS-CoV-2 lineage, clade, or SNV frequency across multiple locations (*Figure 1C*) as we will demonstrate using case studies. COVID-19 CG provides functionalities that are complementary to other existing public browsers (see Discussion) and were designed to empower these specific user groups:

*Vaccine and therapeutics developers* can inform the design and testing of their vaccine, antibody, or small molecule by using COVID-19 CG to rapidly identify all of the variants in their targeted SARS-CoV-2 protein or antigen, alongside the frequency of each variant in their geographical location(s) of interest. Scientists can use COVID-19 CG to generate hypotheses and determine whether the variants present in the location of vaccine/therapeutic implementation may impact their product-specific interaction interface or antigen.

### Case study of SNVs in the receptor binding domain of the SARS-CoV-2 spike

Analyzing SNVs by geography and time is critical as the frequency of each SNV may vary significantly across different regions over time. For instance, as of December 2020, an S477N mutation in the receptor binding domain (RBD) has become dominant in Australia (69% of Australian SARS-CoV-2 genotypes, all time) although it constitutes less than 6% of SARS-CoV-2 genotypes globally (*Figure 2A*). SNV frequency in a given region can also shift over time, for example, an RBD N439K mutation not found in Ireland prior to July was present in 42% of the genomes collected mid-July through September, peaking in August and gradually fading after (*Figure 2B*). Another rare RBD

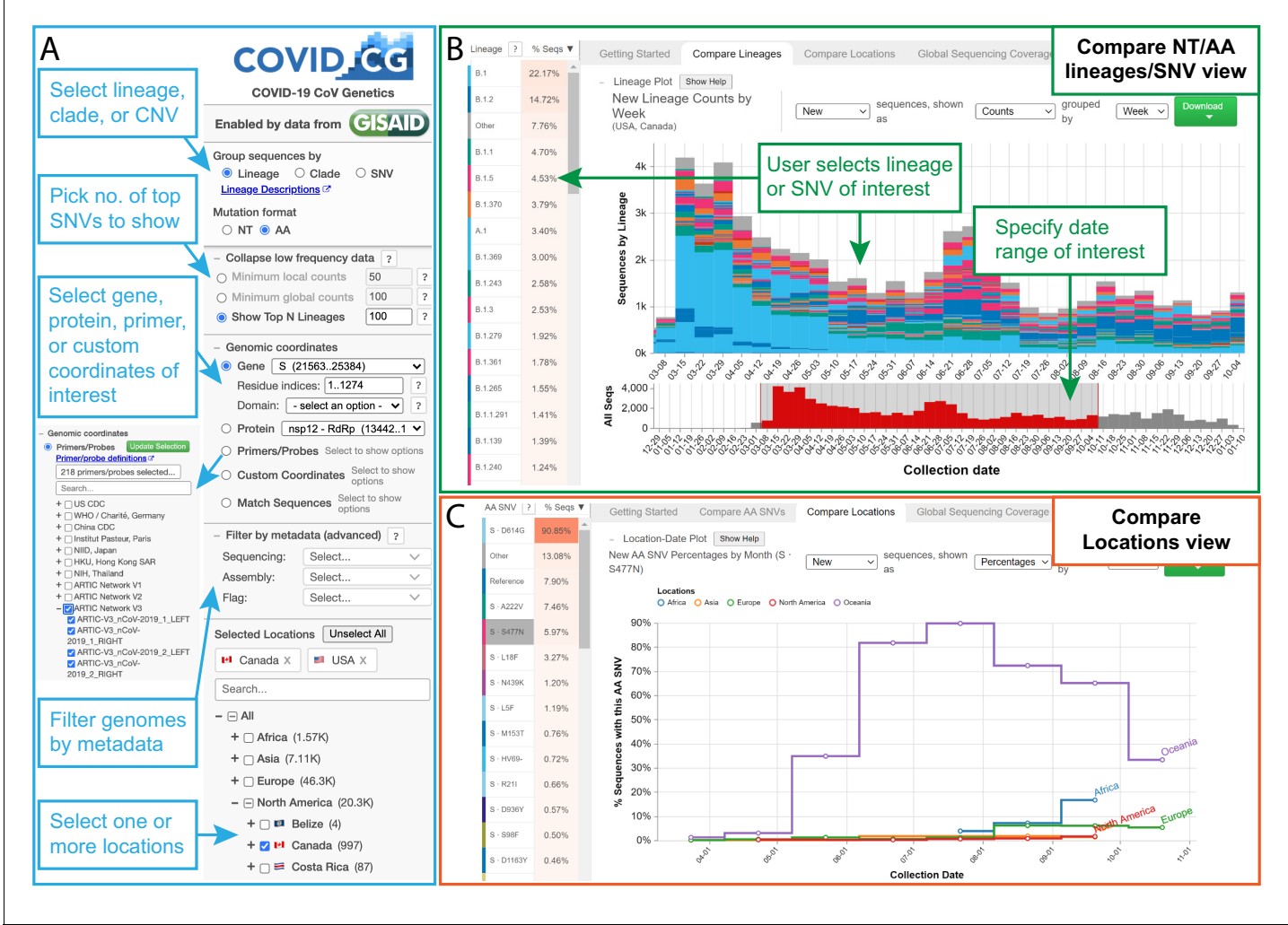

**Figure 1.** The COVID-19 CG (https://covidcg.org) browser interface. (**A**) Users can select SARS-CoV-2 genomes according to lineage, clade, or single-nucleotide variation (SNV), virus gene or protein, and location(s). Genomes can also be filtered by metadata and specifically analyzed at genomic coordinates of interest, such as the target sites of 665 commonly used diagnostic primers and probes. (**B**) In the 'Compare lineages/clades/SNVs' tab, users can visualize SARS-CoV-2 lineages, clades, or SNVs by location, define their date range of interest, and see the corresponding SNVs at the nucleotide or amino acid level. Lineage/clades/SNVs can be sorted by frequency and name/position. (**C**) In the 'Compare locations' tab, users can compare the frequencies of specific lineages, clades, or SNVs in multiple locations over time.

The online version of this article includes the following figure supplement(s) for figure 1:

**Figure supplement 1.** The number of global SARS-CoV-2 genome sequences and mutations is accumulating.

**Figure supplement 2.** COVID-19 CG computational workflow.

S477N mutation, which was found in only 1% of the Australian SARS-CoV-2 sequences before June, has constituted 84% of the sequenced June through December genomes (*Figure 2C*). This geographical and temporal variation is important to incorporate into the design and testing of therapeutic antibodies (such as those under development as therapeutics by Regeneron that specifically target the SARS-CoV-2 spike RBD), as well as mRNA or recombinant protein-based vaccines. This will help to assure developers of the efficacy of their therapeutics and vaccines against the SARS-CoV-2 variants that are present in the intended location of implementation.

In addition, COVID-19 CG can be harnessed to track changes in SARS-CoV-2 evolution post-implementation of therapeutics and vaccines. It will be crucial to watch for rare escape variants that could resist drug- or immune-based interventions to eventually become the dominant SARS-CoV-2 variant in the community. This need was particularly emphasized by a Regeneron study that

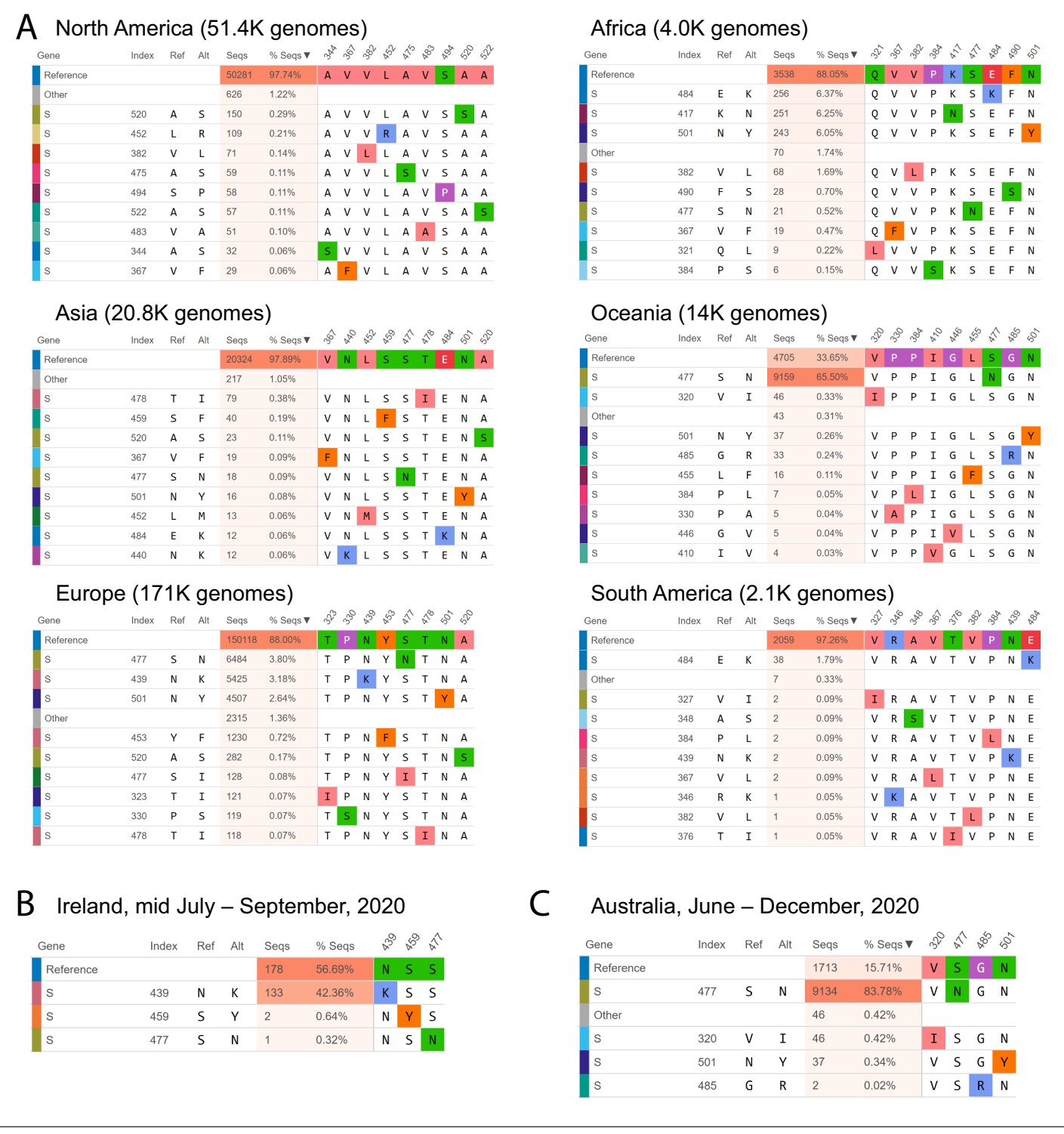

**Figure 2.** Mutational frequencies in the SARS-CoV-2 spike receptor binding domain (RBD) across geographical location and time. Screen captures from the Compare AA SNVs tab on December 29, 2020, are shown. (**A**) The top 10 RBD SNVs alongside the number of high quality sequences available on GISAID are shown for each region. (**B**) The top RBD SNVs for Ireland between mid-July and August 2020 are shown. The S439N mutant had not been previously detected in Ireland. (**C**) The top RBD SNVs for Australia between June and December 2020 are shown. The S477N mutant constituted only 1.05% of the Australian SARS-CoV-2 genomes on GISAID prior to June.

demonstrated that single amino acid variants could evolve rapidly in the SARS-CoV-2 spike to ablate binding to antibodies that had been previously selected for their ability to neutralize all known RBD variants; some of these RBD amino acid variations are already present at low frequency among human SARS-CoV-2 genomes globally (*Baum et al., 2020*). The authors, Baum et al., suggested that these rare escape variants could be selected under the pressure of single antibody treatment, and, therefore, advocated for the application of cocktails of antibodies that bind to different epitopes to minimize SARS-CoV-2 mutational escape. A recent study by *Greaney et al., 2021* generated high-resolution 'escape maps' delineating RBD mutations that could potentially result in virus escape from neutralization by 10 different human antibodies. Based on lessons learnt from the rise of multi-drug resistant bacteria and cancer cells, it will be of the utmost importance to continue tracking SARS-CoV-2 evolution even when multiple vaccines and therapeutics are implemented in a given human population.

*Diagnostics developers* can evaluate their probe, primer, or point-of-care diagnostic according to user-defined regional and temporal SARS-CoV-2 genomic variation. More than 665 established primers/probes are built into COVID-19 CG, and new diagnostics will be continually incorporated into the browser as they become publicly available. Users can also input custom coordinates or sequences to evaluate their own target sequences and design new diagnostics.

## Case study of SNVs that could impact the sensitivity of diagnostic primers

A recent preprint alerted us to the finding that a common G29140T SNV, found in 22.3% of the study's samples from Madera County, California, was adversely affecting SARS-CoV-2 detection by the NIID_2019-nCoV_N_F2 diagnostic primer used at their sequencing center; the single SNV caused a ~ 30-fold drop in the quantity of amplicon produced by the NIID_2019-nCov_N_F2/R2 primer pair (*Vanaerschot et al., 2020*). We used COVID-19 CG to detect other SNVs that could impact the use of this primer pair, discovering that there are SARS-CoV-2 variants in several countries with a different C29144T mutation at the very 3' end of the same NIID_2019-nCoV_N_F2 primer (*Figure 3A*). The authors of the preprint, Vanaerschot et al. noted that SNVs could impact assay accuracy if diagnostic primers and probes are also being used to quantify viral loads in patients. We found that at least 10 other primer pairs could potentially be at risk in different geographical regions due to SNVs that appear proximal to the 3' ends of primers (*Figure 3B–K*): China-CDC-N-F and R; NIH, Thailand, WH-NIC N-F; US CDC 2019-nCoV-N1-R; US CDC 2019-nCoV-N2-F; ARTIC-V3_nCoV-2019_11_RIGHT; ARTIC-V3_nCoV-2019_13_LEFT; ARTIC-V3_nCoV-2019_34_LEFT; ARTIC-V3_nCoV-2019_39_LEFT (note that the ARTIC primers are used for nanopore sequencing; *Tyson et al., 2020*); WHO N_Sarbarco_R1; and Institut Pasteur, Paris 12759Rv. Labs and clinics can use COVID-19 CG (https://covidcg.org) to check their most commonly used primers and probes against the SARS-CoV-2 sequences that are prevalent in their geographical regions. We reiterate Vanaerschot et al.'s exhortation that SARS-CoV-2 detection strategies should ideally target multiple viral genes and check for concordant Ct values. In addition, scientists have cautioned against lab-specific errors that exist among the available SARS-CoV-2 genomes, often co-localizing with commonly used primer binding sites (*Turakhia et al., 2020*). These errors appear to be specific to particular labs and do not affect all of the genomes on the GISAID database. Therefore, we have added a caution to our site and a link to the evolving list of problematic sites (*Issues with SARS-CoV-2 sequencing data, 2020*) so that users are made aware of these potential errors in some subsets of the genomes deposited on GISAID.

*Researchers and public health professionals* can use COVID-19 CG to gain insights as to how the virus is evolving in a given population over time (e.g., in which genes are mutations occurring, and do these lead to structural or phenotypic changes?). For example, users can track D614G distributions across any region of interest over time. *Figure 4* shows a variety of different D614G population dynamics in different areas. Nonetheless, we caution against inferring (1) chains or directionality of transmission and (2) changes in the transmissibility of any SARS-CoV-2 SNV based on population dynamics alone. Inconsistent sampling, sampling biases, differences in founder host population traits (even median patient age), superspreading events, regionally and temporally differential travel restrictions, and numerous other factors instead of virus biological differences can influence the global distribution of SNVs (*Grubaugh et al., 2020*). Our site carries the following warning:

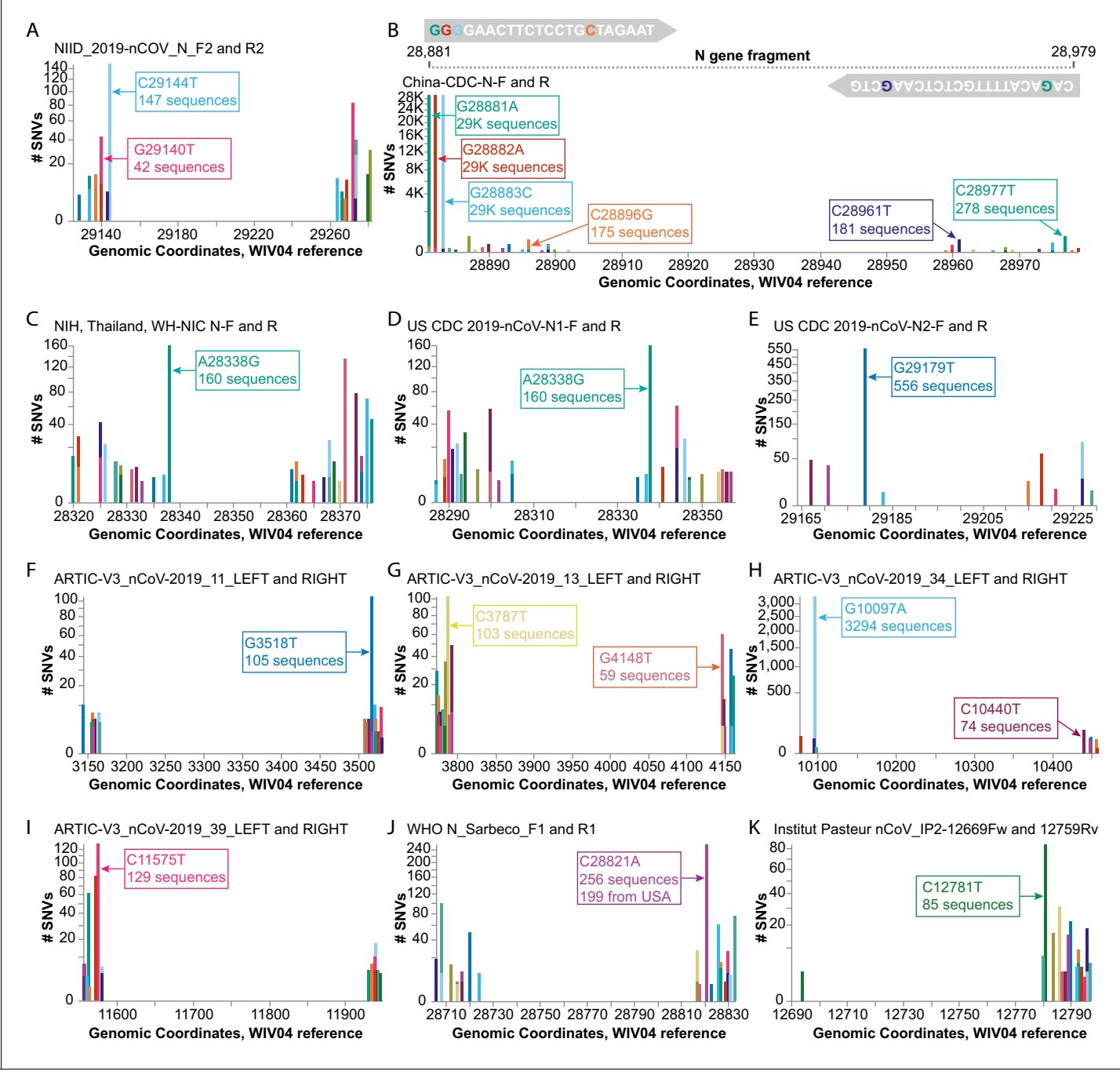

**Figure 3.** Investigating diagnostic-targeted regions of the SARS-CoV-2 genome for single-nucleotide variations (SNVs) that could impact primer/probe sensitivity. Images were downloaded from the Compare NT SNVs tab in September 2020. Labels for specific mutations were added. Primer pairs that contain at least one primer with potentially impactful SNVs near the 3' end are shown. None of the 11 primer pairs shown here were designed with degenerate bases. (**A**) The G29140T mutation has been demonstrated to impact the NIID_2019-nCOV_N_F2 primer sensitivity. (**B-K**) Primer pairs affected by SNVs with a global frequency of more than 80 instances are shown. (**B**) As an example, majors SNVs are colored accordingly in the China-CDC-N-F and R (forward and reverse) primers.

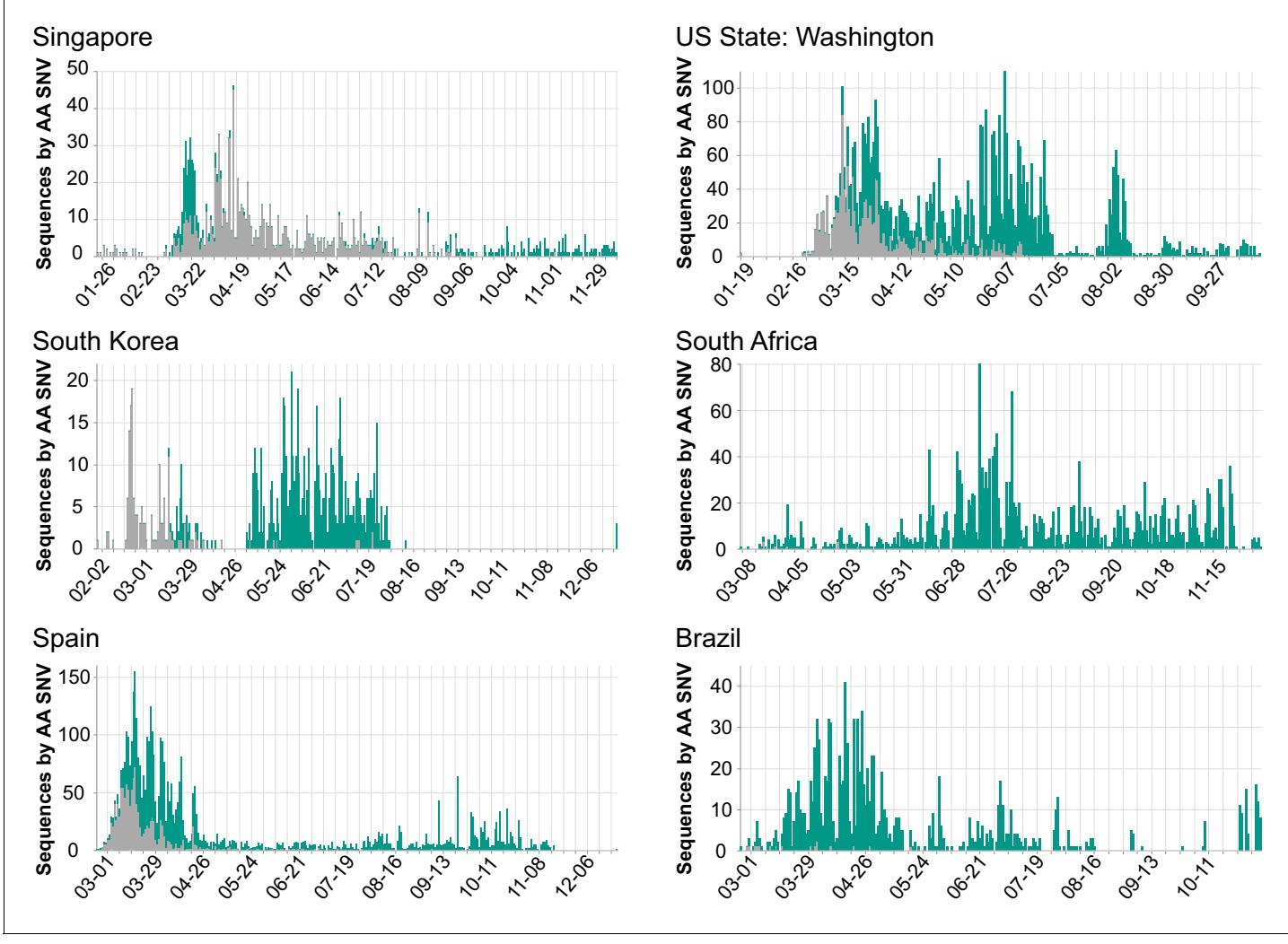

**Figure 4.** Population dynamics of spike D614G in different regions. Images were downloaded from the Compare Lineages tab of covidcg.org on December 29, 2020: The spike D614 variants are shown in gray, and the G614 variants are shown in green. Plots displaying different population dynamics were deliberately selected. Time is shown on the horizontal axis and the number of sequences is shown on the vertical axis; these differ per country depending on when and how many samples were collected and whether the sequences were deposited onto GISAID by December 29, 2020.

'Inconsistent sampling in the underlying data can result in missing data and artefacts in this visualization. Please interpret this data with care.'

## Case study of Australia's new dominant SARS-CoV-2 variant

In September 2020, we observed that the SARS-CoV-2 spike S477N mutation had become more prevalent in Australia (*Figure 5A*). Globally, the S477N mutation was first detected in a single sample of lineage B.1.1.25 that was collected on March 19, 2020, in Victoria, Australia, and became the dominant SARS-CoV-2 variant in the region between June and September (*Figure 5B*). In particular, the set of SNVs that co-occur with the S477N mutation in Australia (all time, as well as prior to May 2020 before the most recent outbreak) are different from the set of co-occurring SNVs in the United Kingdom (*Figure 5C*) — suggesting that the S477N mutation occurred separately in the Australian and the UK lineages. However, COVID-19 CG only reflects data contributed to GISAID. Variants of interest could be present in other countries, but not yet known to the public because the sequencing centers in those countries have not collected or deposited their data in GISAID. Furthermore, in instances where only a singular, sporadic variant is detected (no sustained transmission), there is also the possibility of sequencing error resulting in incorrect lineage assignment. Due to these caveats,

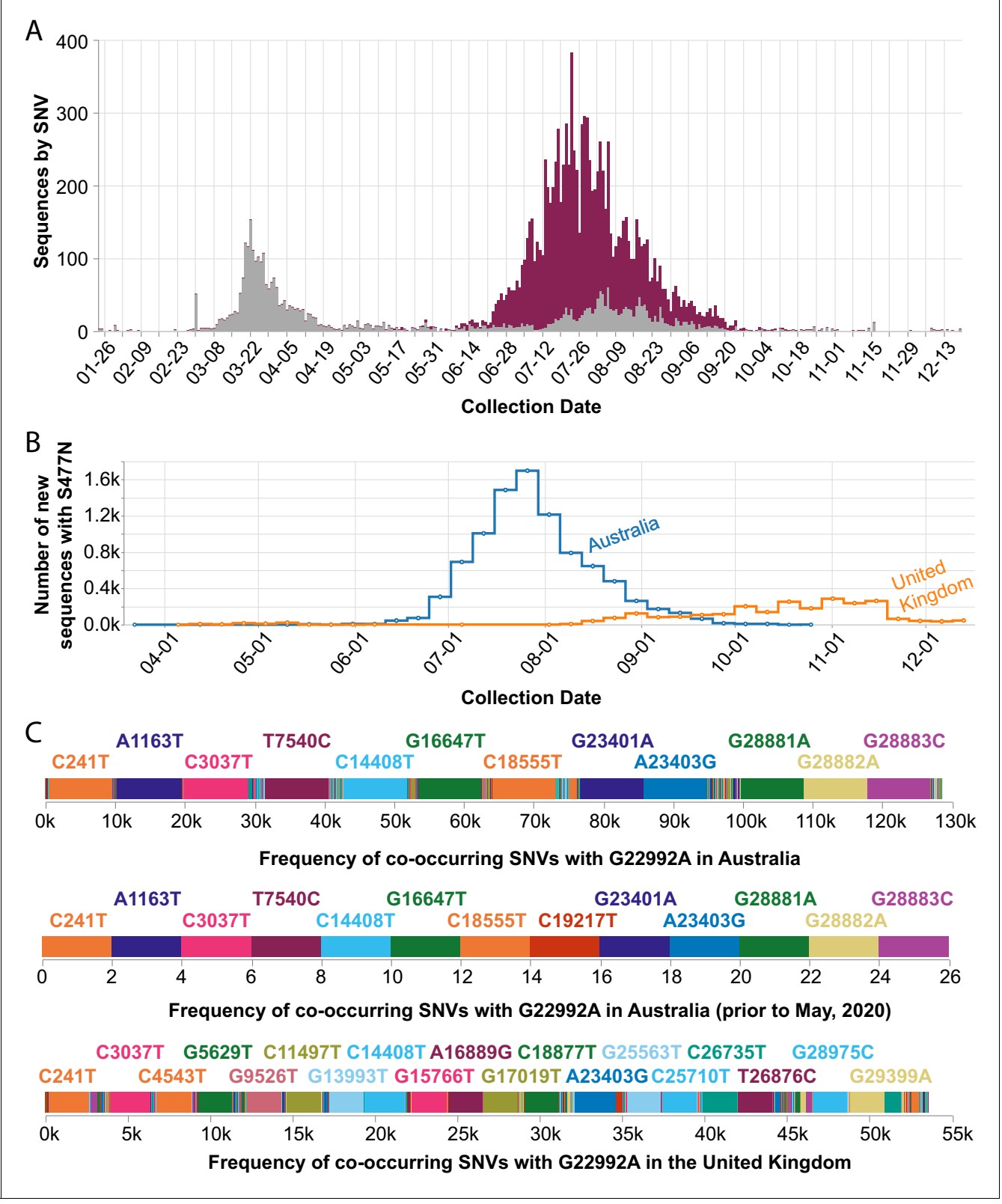

**Figure 5.** Frequency of the spike S477N mutation in Australia over time. (**A**) Image downloaded from the Compare SNVs tab of covidcg.org on December 29, 2020: SARS-CoV-2 variants bearing the spike S477N mutation (also known as the G22992A SNV; depicted in purple), the majority of which lie in the B.1.1.25 lineage, had become the most prevalent form of SARS-CoV-2 in Australia in June through September 2020. (**B**) Image downloaded from the Compare Locations tab of covidcg.org on December 29, 2020: the cumulative percent of sequences carrying the S477N mutation

*Figure 5 continued on next page*

Figure 5 continued

in Australia and the United Kingdom. (C) Images downloaded from the Compare NT SNVs tab of covidcg.org on December 29, 2020: co-occurring SNVs of G22992A (spike S477N) in Australia, all time versus prior to May 2020, versus in the United Kingdom. In addition to the graphical view, co-occurring mutation data is downloadable in an image or table format.

the genetic data must be used in combination with other types of data, such as from contact tracing efforts, before it is possible to draw conclusions about the international transmission of SARS-CoV-2 variants. In the case of the S477N variant that is now dominating in Australia, the sequencing data alone indicate that the local transmission of this variant in Australia since March 2020 or earlier cannot be ruled out.

## Discussion

COVID-19 CG is one of a growing number of COVID-19 public browsers that analyze and visualize the SARS-CoV-2 genomes in the GISAID database, serving different and complementary user objectives. NextStrain (*Hadfield et al., 2018*) visualizes real-time tracking of SARS-CoV-2 evolution on a global and continental level, using phylogenetic trees, geographical maps, and entropy and frequency plots; CoV-GLUE (*Singer et al., 2020*) is a browsable database of amino acid replacements and indels; the UCSC SARS-CoV-2 Genome Browser is an extension of their widely used browser that enables layering of annotation tracks and new features such as conservation with similar viruses, immune epitopes, primers, and CRISPR guides (*Fernandes et al., 2020*); the COVID-19 Viral Genome Analysis Pipeline (*Korber et al., 2020*) allows for exploring mutations geographically and over time with a focus on the spike protein; COVIDep (*Ahmed et al., 2020*) is a browser for real-time reporting of vaccine target recommendations; Genomic Signature Analysis of Virus Causing COVID-19 (*Bauer et al., 2020*) visualizes the similarity between SARS-CoV-2 genomes in 2D space; SARS-CoV-2 Alignment Screen (*van Dorp et al., 2020a*; *van Dorp et al., 2020b*) is a visualization tool providing the distribution of SNPs and homoplasies; the WashU Virus Genome Browser (*Flynn et al., 2020*) provides different tools such as a phylogenetic tree and a genomic-coordinate track-based view of viral sequencing data. There are also websites aimed at helping users to check COVID-19 diagnostic primers and probes, for example, the SARS-CoV-2 target regions browser from the European Commission and the Status of SARS-CoV-2 detection systems (RT-PCR) browser from the University of Turin. GISAID also monitors and reports RBD mutations and assigns clades and lineages as well as genome quality. The current GISAID reports describe the geographical distribution of new mutations, total occurrence by country, mutations in commonly used diagnostics, and 3D structural maps of mutations in the four largest clades (gisaid.org/spike). GISAID additionally provides CoVsurver (gisaid.org/covsurver) for users to screen for potential phenotypic changes based on curated literature annotations; BLAST searches and large-scale phylogenetic trees and alignments; and a new functionality for tracking emerging variants (gisaid.org/variants).

COVID-19 CG provides additional complementary functionalities to those that are available on the GISAID platform, as well as these other diverse COVID-19 public browsers. COVID-19 CG (https://covidcg.org) was built to help scientists and professionals worldwide, with varying levels of bioinformatics expertise, in their real-time analysis of SARS-CoV-2 genetic data. Mainly, COVID-19 CG enables users to track in real time, without sub-sampling, lineages, clades, and SNVs (nucleotide or amino acid) across the SARS-CoV-2 genome (or any genomic region of choice) while rapidly filtering on a user-friendly interface by geographical regions, date range, and other criteria of interest to the user. In this work, we explored several case studies that serve to highlight what users can quickly achieve using COVID-19 CG. As more detailed metadata is generated by COVID-19 studies and initiatives, we will update the application to enable filtering according to patient traits such as host species (e.g., human or mink), gender, age, ethnicity, and medical condition (e.g., symptoms, hospitalization). In addition, over the next year, as more mutations accumulate across the tens of millions of COVID cases worldwide, we plan to implement a server-client model for COVID-19 CG, where genomic data is filtered and processed on the server before being sent to the client for visualization. This change should significantly reduce the computational burden of COVID-19 CG on user's computers, and allow our application to scale to a much larger number of genomes. Our team and colleagues continue to actively use the COVID-19 CG site to quickly generate hypotheses about

COVID-19 before performing a deep analysis using the data on GISAID. We anticipate that novel questions will arise over the next year and intentionally designed COVID-19 CG (https://covidcg.org) to be modular in order to continually integrate different types of COVID-19 data and build in new features. Some of these functionalities exist on other platforms, and we recommend that scientists try each browser to find the one that best meets their needs.

SARS-CoV-2 public browsers such as COVID-19 CG and others we have highlighted here help scientists around the world to parse through large quantities of SARS-CoV-2 genomic data and metadata for the purposes of informing vaccine, therapeutics, and policy development. We advocate for decision makers around the world to sustain or accelerate their sequencing of virus genomes in their geographical area. Collecting virus genomic data is particularly relevant to regions that are experiencing increases in COVID-19 cases. If only sparse genomic data are sampled, we risk the late detection of SARS-CoV-2 variants that exhibit enhanced transmissibility, virulence or resistance against therapeutics or vaccination programs in these pandemic hotspots. Furthermore, the widespread implementation of vaccines and antibody therapies could stimulate the emergence and selection of new escape variants (*Baum et al., 2020*). To compound these risks, SARS-CoV-2 transmission from humans to minks (and in some cases back into humans) has already been detected at farms across at least 10 countries and one wild mink in the United States (*OIE - World Organisation for Animal Health, 2020*). This process of species crossing, if left unchecked, can risk the spread of SARS-CoV-2 into wild animal populations and the emergence of diverse SARS-CoV-2 variants.

Coordinated sequencing and contact tracing efforts (e.g., in the United Kingdom, Singapore, the Netherlands, Italy, California, and Australia) emphasize the urgency of establishing open access platforms to evaluate trends in virus introduction into each country or region in real time. Local policymakers, public health researchers, and scientists can use global SARS-CoV-2 genetic data, in complementation with contact tracing data, to better understand which lineages were imported into their region (from which potential international locations), whether these were introduced multiple times, and if particular lineages are dying out or persisting. Labs in numerous countries are already making incredible efforts to sequence the SARS-CoV-2 variants circulating in their local populations (*Figure 6*). When each country actively contributes to the database of SARS-CoV-2 genomes, this protects against sampling biases that can impact the ability to perform phylogenetic analysis and interpret global SARS-CoV-2 data. Toward this goal that affects all of humanity, we advocate for the increased sequencing of SARS-CoV-2 isolates from patients (and infected animals) around the world, and for these data to be shared in as timely a manner as possible.

## Materials and methods

### Data pipeline

Our data processing pipeline is written with the Snakemake scalable bioinformatics workflow engine (*Köster and Rahmann, 2012*), which modularizes our workflow and enables reproducibility and compatibility with cloud-computing. All code and relevant documentation are hosted on an open-source, publicly available GitHub repository (https://github.com/vector-engineering/COVID19-CG; *Chen, 2021*; copy archived at swh:1:rev:e9558dc11b31b908f3af142e403d33e91d417b8a), providing example data for users to validate our pipeline.

Data analysis is broken up into two Snakemake pipelines: (1) ingestion and (2) main. The ingestion pipeline downloads, chunks, and prepares metadata for the main analysis, and the main pipeline analyzes sequences, extracts SNVs, and compiles data for display in the web application. Configuration of the pipeline and the web application is defined by a single YAML file.

### Data ingestion

At the time of publication, two ingestion workflows are available: *workflow_genbank_ingest* and *workflow_gisaid_ingest*. While the GISAID ingestion pipeline is provided as open source, it utilizes a custom GISAID endpoint and is intended only for internal use. Both ingestion pipelines are designed to be run daily and chunk sequence data in order to minimize expensive reprocessing/realignment in the downstream main analysis step. In addition, these ingest pipelines clean the provided sequence metadata by removing unwanted fields, renaming fields, and fixing typos (see 'Metadata Cleaning' for details).

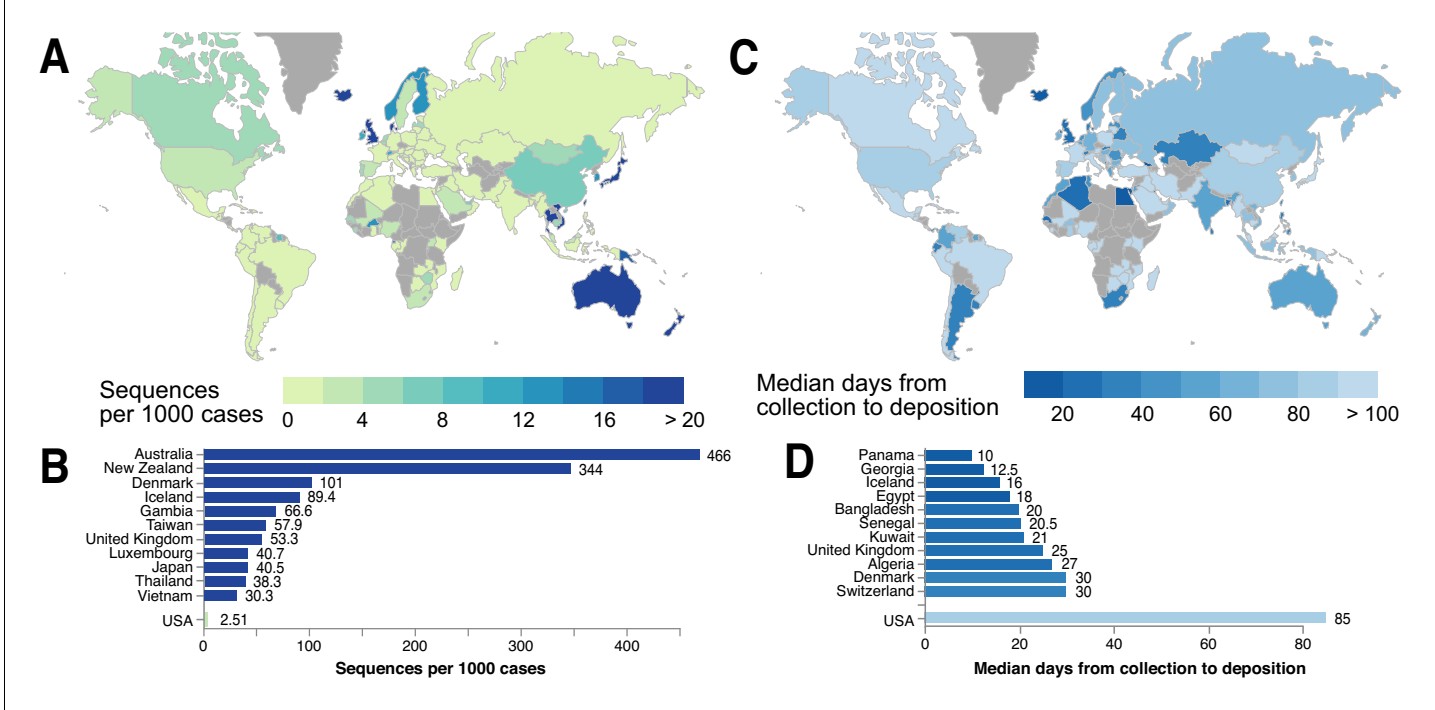

**Figure 6.** Global statistics of SARS-CoV-2 genome data contribution to GISAID. Interactive and more comprehensive versions of the figure panels are displayed on the Global Sequencing Coverage tab of covidcg.org. (A) A world map of countries labeled by the number of SARS-CoV-2 sequences contributed per 1000 cases. (B) A bar graph showing the sequences per 1000 cases for the top five countries and the United States. Countries with less than 500 cases were excluded from this plot. (C) A world map of countries labeled by median days between sample collection and sequence deposition. (D) A bar graph showing the median days from collection to deposition for the top five countries and the United States. These interactive displays are generated using sequencing data from GISAID and case data from the JHU CSSE COVID-19 Dataset (*Dong et al., 2020*).

## Metadata cleaning

We clean metadata with the aim of preserving the original intent of the authors and data submitters while presenting simpler and unified versions to end users. Sequencing metadata is cleaned to remove obvious typos, and to unify labels with the same meaning, for example, 'MinION' and 'Nanopore MinION.' Location metadata is cleaned with the goal of simplifying the location selector in the sidebar. Locations with excessive children are collapsed to the nearest upper hierarchical grouping. For example, if a state has individual data for 200+ towns, these towns will be collapsed to the county level in order to facilitate easier data browsing. Typos and clear identities are also unified to prevent the display of duplicate locations in the application.

## Sequence preprocessing

Based on best practices, we filter out sequences meeting any of the following criteria: (1) present on the NextStrain's exclusion list (https://github.com/nextstrain/ncov/blob/master/defaults/exclude.txt; *Hadfield et al., 2018*), (2) genomes from non-humans (animals, environmental samples, etc.), (3) genome length less than 29,700 nt, or (4) >5% ambiguous base calls. Sequences which pass all pre-processing filters are carried onto the next steps.

## SNV assignments

SNVs and insertions/deletions (indels) at the nucleotide and amino acid level are determined using *bowtie2* by aligning each sequence to the WIV04 reference sequence (WIV04 is a high quality December 2019 genome that is 100% identical to the other commonly used publicly available SARS-CoV-2 genome reference Wuhan-Hu-1/NC_045512.2, excepting the length of the poly(A) tail). Spurious SNVs and probable sequencing errors, defined as less than three global occurrences, are filtered out prior to downstream analysis. SNVs involving ambiguous base calls ('N' in the original

sequences) are ignored. Indels resulting in frameshifts are ignored, and SNVs/indels occurring in non-protein-coding regions are ignored when determining SNVs/indels on the AA level.

### Lineage/clade analysis

Viral lineages, as defined by the *pangolin* tool (*Rambaut et al., 2020*), and clades (*Tang et al., 2020*) are provided by GISAID. In accordance with *pangolin*, SNVs present in >90% of sequences within each lineage/clade will be assigned as lineage/clade-defining SNVs.

### Application compilation

The web application is written in Javascript and primarily uses the libraries React.js, MobX, and Vega. The code is compiled into Javascript bundles by webpack. Sequence data and metadata are combined and compressed into one file, which is downloaded by the web application on page load. This allows us to regularly update the data on https://covidcg.org.

### Application deployment

COVID CG (https://covidcg.org) is hosted by Google Cloud Run. The application code is assembled into a Docker image (see Dockerfile), with a build environment (node.js) and deployment environment (NGINX).

### Data availability

All of the data shown in this manuscript and displayed on COVID CG (https://covidcg.org) are downloaded from the GISAID EpiCov database (https://www.gisaid.org). All code and relevant documentation are hosted on an open-source, publicly available GitHub repository (https://github.com/vector-engineering/COVID19-CG).

## Acknowledgements

BD and AC are supported by awards from the National Institute of Neurological Disorders and Stroke (UG3NS111689) and a Brain Initiative award funded through the National Institute of Mental Health (UG3MH120096) and from the Stanley Center for Psychiatric Research. YAC is supported by a Broad Institute SPARC award, a BroadIgnite award, and the Stanley Center for Psychiatric Research. We gratefully acknowledge all of the authors from the originating laboratories responsible for obtaining the specimens and the submitting laboratories where genetic sequence data were generated and shared via the GISAID Initiative (*Supplementary file 1* for all contributors; *Supplementary file 2* for all Australian contributors whose data were used to generate *Figure 5*). We also thank the GISAID team for their critical reading of the manuscript and helpful feedback.

## Additional information

#### Competing interests

Shing Hei Zhan: is a current employee and shareholder of Fusion Genomics Corporation, which develops molecular diagnostic assays for infectious diseases including COVID-19. The other authors declare that no competing interests exist.

#### Funding

| Funder | Grant reference number | Author |
|---|---|---|
| National Institute of Neurological Disorders and Stroke | UG3NS111689 | Albert Tian Chen<br>Yujia Alina Chan<br>Benjamin E Deverman |
| National Institute of Mental Health | UG3MH120096 | Albert Tian Chen<br>Benjamin E Deverman |
| Stanley Center for Psychiatric Research, Broad Institute | | Albert Tian Chen<br>Yujia Alina Chan<br>Benjamin E Deverman |

The funders had no role in study design, data collection and interpretation, or the decision to submit the work for publication.

#### Author contributions
Albert Tian Chen, Conceptualization, Data curation, Software, Formal analysis, Investigation, Visualization, Methodology, Writing - original draft, Writing - review and editing; Kevin Altschuler, Software, Visualization, Methodology, Writing - review and editing; Shing Hei Zhan, Conceptualization, Formal analysis, Investigation, Methodology, Writing - review and editing; Yujia Alina Chan, Conceptualization, Investigation, Visualization, Methodology, Writing - original draft, Writing - review and editing; Benjamin E Deverman, Conceptualization, Resources, Supervision, Investigation, Visualization, Methodology, Writing - original draft, Project administration, Writing - review and editing

#### Author ORCIDs
Yujia Alina Chan (iD) https://orcid.org/0000-0002-0731-637X
Benjamin E Deverman (iD) https://orcid.org/0000-0002-6223-9303

#### Decision letter and Author response
Decision letter https://doi.org/10.7554/eLife.63409.sa1
Author response https://doi.org/10.7554/eLife.63409.sa2

## Additional files

#### Supplementary files
- Supplementary file 1. List of SARS-CoV-2 genome contributors to GISAID Initiative.
- Supplementary file 2. List of Australian SARS-CoV-2 genome contributors to GISAID Initiative.
- Transparent reporting form

#### Data availability
All of the data shown in this manuscript and displayed on COVID CG (https://covidcg.org) are downloaded from the GISAID EpiCovTM database (https://www.gisaid.org). All code and relevant documentation are hosted on an open-source, publicly available GitHub repository (https://github.com/vector-engineering/COVID19-CG; copy archived at https://archive.softwareheritage.org/swh:1:rev:e9558dc11b31b908f3af142e403d33e91d417b8a/).

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
