## [Decision Letter]

**Acceptance summary:**

This publication describes covidcg.org, which is a useful interface for tracking and visualizing mutations that have appeared in sequenced SARS-CoV-2 isolates. It provides complementary functionality to a variety of other interfaces, and together these platforms aid in the important task of tracking the emergence and potential impacts of mutations in SARS-CoV-2.

**Decision letter after peer review:**

Thank you for submitting your article "COVID-19 CG enables SARS-CoV-2 mutation and lineage tracking by locations and dates of interest" for consideration by *eLife*. Your article has been reviewed by two peer reviewers, one of whom is a member of our Board of Reviewing Editors, and the evaluation has been overseen by Miles Davenport as the Senior Editor. The reviewers have opted to remain anonymous.

The reviewers have discussed the reviews with one another and the Reviewing Editor has drafted this decision to help you prepare a revised submission.

Summary:

Devermen and colleagues describe COVID-19 CG, a resource for information on SARS-CoV-2 data. They present several use cases demonstrating the utility of this resource. Given the nature of the pandemic, integration of datasets in easy to find and analyzable formats are of significant importance. While many other tools provide similar functionalities, covidcg is clearly useful to researchers and likely easier to use and more intuitive in specific use cases. Indeed, we note that one of us has made use of it for the specific purpose of tracking mutations by geographic location.

Essential revisions:

We had the following suggestions that we think you should find easy to address in a revised version:

The current interface / setup seems designed for the situation when there are relatively few mutation (e.g., you get a list of all mutations in a region and click on mutation of interest). This seems like it will become increasingly unwieldy as more and more mutations reach appreciable frequencies as the virus evolves. What is the plan to deal with this? Some mention of this point might be helpful in the Discussion.

The Abstract should have a link to the website: https://covidcg.org. This will make the resource easier to find and allow users to interact with it while reading the paper.

The authors should note many of the existing sequence resources available including: the UCSC SARS-CoV-2 browser (https://genome.ucsc.edu/covid19.html) , the Wash U Genome browser (https://virusgateway.wustl.edu/), Nextstrain (https://nextstrain.org/) as well as the COG-UK efforts (https://www.cogconsortium.uk/, including CoV-GLUE and Pangolin which is cited, but only in the Materials and methods). These other tools are not competitors but rather complementary and researchers should be encouraged to use the tool that is easiest for them to use and makes the most sense for a specific analysis. Thus "cannot be found in other public browsers" should be reworded.

Suggested citations:

https://www.nature.com/articles/s41588-020-0700-8

https://www.nature.com/articles/s41588-020-0697-z

https://europepmc.org/article/ppr/ppr177298

https://academic.oup.com/bioinformatics/article/34/23/4121/5001388

Related to the above point, overall this is a useful resource, but the authors should be clear that many other resources are available and that the scientific community should use the best tools for a particular task to combat this pandemic. In that respect, any additional discussion weighing the strengths and uses of various above resources would probably be useful.

With regard to the mutations that might disrupt primer binding sites the authors should compare them to problematic/masked sites that are almost certainly due to sequencing/assembly error (e.g. within ARCTIC primers) as they do not obey expected evolutionary patterns. A full list has been collected by the community here: https://github.com/W-L/ProblematicSites_SARS-CoV2/ and is discussed in this paper (https://journals.plos.org/plosgenetics/article?id=10.1371/journal.pgen.1009175); see also here (https://virological.org/t/issues-with-sars-cov-2-sequencing-data/473/14). The authors should, in addition to their recommendation that scientists check mutation rates at primer binding sites, also reiterate Vanaerschot et al.'s suggestion to interrogate multiple target genes and compare Ct values.

Subsection “Case study of SNVs in the receptor binding domain (RBD) of the SARS-CoV-2 Spike”: Although Baum et al. identified mutations by deep sequencing both in and out of the RBD, they only validated that mutations in the RBD affected binding by Regeneron antibodies. The others are likely hitchhikers, and should not be referred to as possible escape mutants in the absence of direct functional testing

---

## [Author Response]

Essential revisions:We had the following suggestions that we think you should find easy to address in a revised version:The current interface / setup seems designed for the situation when there are relatively few mutation (e.g., you get a list of all mutations in a region and click on mutation of interest). This seems like it will become increasingly unwieldy as more and more mutations reach appreciable frequencies as the virus evolves. What is the plan to deal with this? Some mention of this point might be helpful in the Discussion.

The ever increasing volume of SARS-CoV-2 genomes is a significant challenge for many COVID-19 browsers. Currently, SNV data and metadata for all 300K+ SARS-CoV-2 genomes are sent directly to the browser and processed on the front-end, but we understand that this strategy is not sustainable in the long-term. Therefore, we plan to implement a server-client model for COVID-19 CG, where genomic data is filtered and processed on the server before being sent to the client for visualization. This change should significantly reduce the computational burden of COVID-19 CG on user’s computers, and allow our application to scale to a much larger number of genomes.

The Abstract should have a link to the website: https://covidcg.org. This will make the resource easier to find and allow users to interact with it while reading the paper.

Thank you very much for this suggestion. We have added the url to our Abstract. We have also made our Abstract more concise to meet the *eLife* 150-word limit.

The authors should note many of the existing sequence resources available including: the UCSC SARS-CoV-2 browser (https://genome.ucsc.edu/covid19.html) , the Wash U Genome browser (https://virusgateway.wustl.edu/), Nextstrain (https://nextstrain.org/) as well as the COG-UK efforts (https://www.cogconsortium.uk/, including CoV-GLUE and Pangolin which is cited, but only in the Materials and methods). These other tools are not competitors but rather complementary and researchers should be encouraged to use the tool that is easiest for them to use and makes the most sense for a specific analysis. Thus "cannot be found in other public browsers" should be reworded.Suggested citations:https://www.nature.com/articles/s41588-020-0700-8https://www.nature.com/articles/s41588-020-0697-zhttps://europepmc.org/article/ppr/ppr177298https://academic.oup.com/bioinformatics/article/34/23/4121/5001388

We think that the functionalities on COVID-19 CG are largely distinct from the features available on other browsers and are in agreement that these resources are not competing, but complementary to each other and likely useful in combination. For this reason, we have maintained a page of links to other COVID browsers on the “Related Projects” part of the COVID-19 CG page (https://covidcg.org/?tab=related_projects#). We have now reworded that sentence in our manuscript as “are complementary to other existing public browsers (see Discussion)” and expanded our discussion of other browsers in response to the next related point. The four suggested citations have been included in this expanded Discussion.

Related to the above point, overall this is a useful resource, but the authors should be clear that many other resources are available and that the scientific community should use the best tools for a particular task to combat this pandemic. In that respect, any additional discussion weighing the strengths and uses of various above resources would probably be useful.

We agree that it is very important to discuss COVID-19 CG in the context of other resources that are available for different user tasks. We have now highlighted eight of these other resources, on top of the GISAID website, in the first paragraph of our Discussion. A full review that carefully and objectively weighs the strengths and uses of each of the available resources would also be valuable, but is beyond the scope of this article.

With regard to the mutations that might disrupt primer binding sites the authors should compare them to problematic/masked sites that are almost certainly due to sequencing/assembly error (e.g. within ARCTIC primers) as they do not obey expected evolutionary patterns. A full list has been collected by the community here: https://github.com/W-L/ProblematicSites_SARS-CoV2/ and is discussed in this paper (https://journals.plos.org/plosgenetics/article?id=10.1371/journal.pgen.1009175); see also here (https://virological.org/t/issues-with-sars-cov-2-sequencing-data/473/14).

We thank the reviewers for pointing out this issue with SARS-CoV-2 sequencing/assembly. This problem is not straightforward to solve because it deals with lab-specific mutations, and the impact on the analyses on CG is not very clear. As the authors, Turakhia et al. noted, “genuine recurrent mutations that may contain important information about properties of viral evolution are sometimes hard to distinguish from recurrent systematic errors”. They identify numerous lab-associated variants and “recommend their removal prior to phylogenetic tree construction and many subsequent analyses”. Concerningly, this disproportionately affects “variants intersecting or immediately surrounding commonly used primer binding sites”. After much consideration, we believe that this issue should be addressed at the database level or that the specific labs for which this is a problem should be notified and advised on how to amend their data. We had considered masking these sites in the CG browser, but realized that it would throw out data from thousands of genomes that are not impacted by these lab-specific errors. We have added a brief discussion of this issue to the manuscript (in the diagnostics case study section) and have added a new warning to alert users to this issue on covidcg.org.

The authors should, in addition to their recommendation that scientists check mutation rates at primer binding sites, also reiterate Vanaerschot et al.'s suggestion to interrogate multiple target genes and compare Ct values.

We have added a line at the end of the “Case study of SNVs that could impact the sensitivity of diagnostic primers” subsection: “We reiterate Vanaerschot et al.’s exhortation that SARS-CoV-2 detection strategies should ideally target multiple viral genes and check for concordant Ct values.”

Subsection “Case study of SNVs in the receptor binding domain (RBD) of the SARS-CoV-2 Spike”: Although Baum et al. identified mutations by deep sequencing both in and out of the RBD, they only validated that mutations in the RBD affected binding by Regeneron antibodies. The others are likely hitchhikers, and should not be referred to as possible escape mutants in the absence of direct functional testing

We agree with this point and have edited that line to make it clearer and avoid that misunderstanding: “some of these RBD amino acid variations are already present at low frequency among human isolates globally”. Without direct functional testing, we agree that it is difficult to know if a particular mutation is a hitchhiker or artefact of passaging in cells or truly contributing to escape from antibody neutralization.